# Associations of menstrual health with school absenteeism and examination performance among Ugandan secondary school students: A prospective study

Christopher Baleke[1]*, Levicatus Mugenyi[1], Kate A. Nelson[2], Katherine A. Thomas[2], Denis Ndekezi[1], Jonathan Reuben Enomut[1], Connie Alezuyo[3], John Jerrim[4], Helen A. Weiss[2]

1 MRC/UVRI and LSHTM Uganda Research Unit, Kampala, Uganda, 2 International Statistics and Epidemiology Group; Faculty of Epidemiology and Population Health, London School of Hygiene & Tropical Medicine, London, United Kingdom, 3 Ministry of Education and Sports, Kampala, Uganda, 4 UCL Institute of Education, University College London, London, United Kingdom

* Christopher.baleke@lshtm.ac.uk

## Abstract

### Background

Relatively few studies have quantified the amount of school missed due to poor menstrual health, or the impact of poor menstrual health on examination performance.

### Methods

We conducted secondary observational analyses from data nested within a cluster-randomised trial of a menstrual health intervention in 60 Ugandan secondary schools (The trial is registered as ISRCTN45461276). We used baseline data from trial participants in both arms, and endline data from the control arm participants. School absenteeism was estimated as the self-reported number of days absent due to menstruation per month and examination performance was assessed by an independently set assessment by the Uganda National Examination Board. We estimated adjusted incidence rate ratios (aIRR) for associations with school absenteeism, using negative binomial regression adjusted for school-level clustering. We estimated adjusted standardised mean differences (aSMD) in examination scores using mixed-effects linear regression.

### Results

Of the 3312 participants who reported menstruating in the past 6 months at baseline, 323 (9.8%) reported missing at least one day of school per month due to menstruation (mean days missed = 0.30 per month, 95%CI 0.27–0.34). Similarly, of the 1192 participants in the trial control arm seen at endline, 135 (11.3%) reported missing

**Data availability statement:** The datasets used and/or analysed during the current study are available at LSHTM Data Compass and can be freely accessed on reasonable request. Details of location are; MENISCUS Trial quantitative survey data. [Dataset]. London School of Hygiene & Tropical Medicine, London, United Kingdom. 10.17037/DATA.00003865, with the full trial data available at 10.17037/DATA.00003822.

**Funding:** The MENISCUS Trial was supported by the Joint Global Health Scheme with funding from the UK Foreign, Commonwealth and Development Office, the UK Medical Research Council, the UK Department of Health and Social Care through the National Institute of Health Research (NIHR) and Wellcome (grant ref MR/V005634/1).

**Competing interests:** The authors declare that they have no competing interests.

**Abbreviations:** MH, Menstrual health; LMP, Last menstrual period; CI, Confidence interval; LRT, Likelihood ratio test; SMD, Standardized mean difference; IRR, Incidence rate; MENISCUS, Menstrual health interventions, schooling, and mental health problems among Ugandan students; MHM, Menstrual hygiene management; SES, Socio-economic status; WASH, Water, sanitation and hygiene; SAMNS, Self-efficacy in Addressing Menstrual Needs Scale; MHPM, Menstrual Hygiene Preparation and Maintenance sub-scale; MPM, Menstrual Pain Management sub-scale; EST, Executing Stigmatized Tasks sub-scale.

at least one day due to menstruation (mean days missed = 0.31 per month (95%CI 0.27–0.37)). There was evidence that menstrual-related absenteeism and poorer examination performance at endline were both associated with baseline use of inadequate menstrual materials, negative menstrual attitudes, unmet menstrual practice needs, and experience of menstrual-related teasing. In addition, absenteeism due to menstruation was associated with menstrual pain, and poorer examination performance was associated with poorer baseline menstrual knowledge.

## Conclusion

Among Ugandan students, multiple dimensions of menstrual health are associated with school absenteeism and examination performance.

## Trial registration and recruitment details

ISRCTN ISRCTN45461276.

---

## 1. Introduction

Menstrual health (MH) is defined as complete physical, mental, and social wellbeing in relation to the menstrual cycle [1]. Poor menstrual health is prevalent in low- and middle-income countries (LMICs), and may affect educational outcomes such as school absenteeism and examination performance through multiple pathways, including shame, anxiety, lack of confidence, behavioural expectations, poor physical environments, and inadequate menstrual products [2]. A review of 76 qualitative studies, which included 45 from sub-Saharan Africa and 29 from South and East Asia, found that multiple dimensions of menstrual health contributed to school absenteeism and reduced engagement [2]. These dimensions include physical, behavioural and social components such as inadequate menstrual materials, menstrual pain, poor menstrual knowledge and confidence, and inadequate infrastructure for managing menstruation in schools [2].

The statement "one in 10 school-age girls in Africa misses school or drops out for reasons related to her period" is a widely-used statistic, but is not supported by scientific research [3,4]. A systematic review of 15 quantitative studies on school absenteeism during menstruation in Africa found that an average of 31% (95%CI 24–39%) of students reported school absenteeism during menstruation either at last menstrual period (LMP) or over a longer duration [5]. The review found that menstruation also affected concentration, academic performance, and participation in sports [5]. However, the number of school days missed were not quantified and there was a high degree of unexplained heterogeneity between studies ($I^2 = 97.8\%$), with prevalence of absenteeism due to menstrual pain ranging from 6% in Ghana to 65% in Ethiopia. The variation in prevalence may be partly due to variation in the reference period for measurement of school absenteeism, with some using LMP and others using longer durations. A recent multi-country study of data from 673,380 women and girls aged 15–49 in 47 countries found that the proportion reporting menstrual-related school

absenteeism during LMP was 15.0% (95%CI 12.7–17.3%), with the highest absenteeism among 15–19 year olds (17.7%, 95%CI 15.1–20.3%) [6].

There is mixed evidence of an impact of menstrual health interventions on school absenteeism. A recent systematic review identified nine relevant studies in LMICs, of which six found evidence of a positive effect of an intervention on absenteeism [7]. However, the three cluster-randomized control trials included in the review found no effect on absenteeism. Reasons for the lack of evidence on the extent to which menstruation affects school attendance include a lack of standardized and cross-validated absenteeism measures [8]. Additionally these studies lack detail on whether absence was due to menstruation or non-menstrual related factors like general illness or financial inadequacy to education, and the time period over which absence was assessed [9].

A commonly-reported reason for menstrual-related absenteeism is menstrual pain, which can also contribute to reduced participation, concentration, and academic performance [10]. A systematic review of 38 studies (including 23 from LMICs) reported that 20.1% (95% CI 14.9–26.7) of adolescents and young women reported school or university absence due to menstrual pain and 40.9% reporting an adverse effect on classroom performance or concentration [11]. In these reviews, few studies provided details of how the educational outcomes were measured, and none used extant data (official examination results and attendance records) to validate self-reported measures [10]. There is relatively little, and inconclusive evidence on the impact of menstruation on broader educational issues such as self-esteem, participation in school and examination performance [10,11]. Two pilot studies in 6 Ugandan secondary schools showed strong evidence that participants missed school more frequently during menstruation than on other days, based on prospective daily diaries (28.4% of schooldays were missed during menstruation compared with 6.5% when not menstruating; p<0.001 in one study [12], and 15.8% vs 8.6% in the second study [13].

Our recently published MENISCUS cluster-randomised trial ("Menstrual health interventions, schooling, and mental health problems among Ugandan students") showed strong evidence that a multi-component menstrual health intervention improved multiple dimensions of menstrual health [14]. However, the trial showed little evidence of an intervention effect on educational outcomes. The proportion of school days missed during menstruation (measured using prospectively collected daily diaries) was similar by arm (11.2% vs 13.5% in the intervention vs control arms; adjusted odds ratio (aOR)=0.81, 95%CI 0.62, 1.05). Similarly, there was no evidence of an impact of the intervention on examination performance (adjusted mean difference (aSMD)=0.06, 95%CI −0.12, 0.24) [14].

The aim of this paper is to conduct a secondary observational analysis using data from MENISCUS trial [14,15], to estimate associations of menstrual-related experiences with menstrual-related school absenteeism and examination performance, among Ugandan secondary school students. In this paper, we provide new insights by quantifying the associations of menstrual health (including pain management, social support, menstrual practices, and attitudes) with number of days of school absenteeism and examination performance, respectively.

## 2. Methods

### 2.1. Study setting

The MENISCUS trial was conducted in Kalungu and Wakiso districts of Uganda. Wakiso district is the largest and largely urban district in central Uganda with an estimated population of 3.4 million people [16]. In Wakiso, 17% of residents were aged 10–17 years, of whom 49% were attending secondary schools [17]. Kalungu is a rural district with population size 183,232 (2014 census). In Kalungu, 23% were aged 10–17 years, of whom 36% were enrolled in secondary school [18].

### 2.2. Data source

This study is a secondary data analysis of data from the MENISCUS trial. The trial aimed to evaluate whether a multi-component menstrual health intervention improves education, health and wellbeing among female secondary school students in Central Uganda [14]. In this study, we report findings from secondary observational analyses of the baseline

(both arms) and endline (control arm only) trial data. Details of the trial design and results have been published previously [14,15]. In brief, the trial was a parallel-arm, cluster-randomised controlled trial in 60 schools with schools randomised 1:1 to either the intervention package or the control arm. A package of intervention components included puberty education, drama skits, provision of menstrual health kits, training on pain management, and improvements to WASH facilities. The schools randomised to the control arm were provided with a copy of the government guidelines on menstrual hygiene management and sexuality education to the headteacher, and Senior 2 students were given a copy of the government menstrual management reader [15].

### 2.3. Sampling of schools

In preparation for the MENISCUS trial, all government schools and a random selection of private schools were contacted through phone calls and assessed for eligibility using the government's 2019 master list of education institutions. Schools were eligible for the trial if they were mixed-sex, secondary, day- or mixed day/boarding, and had at least minimum water sanitation and hygiene (WASH) facilities (an improved water source and at least one functional sex-specific toilet accessible to female students). Schools also had to have an estimated size of 50–125 or 40–125 female students in Secondary 1 (first year of secondary school) in Wakiso and Kalungu district, respectively. Schools were not eligible for the trial if they were participating in a menstrual health related programme, were exclusively boarding schools, or were single-sex schools. A random sample of 60 schools were selected after being confirmed as eligible and willing to participate. We over-sampled government schools as these are of interest to Ministry of Education stakeholders, and to increase generalizability of findings to these schools.

### 2.4. Participant recruitment and informed consent/assent

All female students enrolled in Secondary 2 (i.e., the second year of secondary school) at the start of the 2022 were eligible to participate in the MENISCUS trial. We sought school-level written consent from headteachers and written informed consent from parents/guardians of students aged <18 years, and from students aged ≥18 years. Students aged <18 years were asked to assent if we had received parental consent. The sample size was fixed by the original trial, and provided 90% power to detect a standardised mean difference of 0.25 between two exposure groups with 1000 students each, assuming a design effect of 1.5.

### 2.5. Data collection

Baseline data were collected from 21 March to 5 July 2022 and endline data were collected from 5 June to 22 August 2023. The surveys included items on sociodemographic characteristics, mental health and multiple dimensions of menstrual health including physical symptoms (e.g., pain), menstrual practices and materials, social experiences (e.g., teasing), and menstrual knowledge and attitudes which were assessed alongside school absenteeism due to menstruation ("menstrual-related absenteeism") or due to other reasons. All were self-completed using Open Data Kit software on tablets. The examinations were independently administered and marked by the Uganda National Examination Board (UNEB), with examinations administered on 16–17 March 2022 (baseline) and 24–26 July 2023 (endline). Baseline examinations comprised materials taught in Secondary 1 and included four mathematics and six biology questions, of which one question was on puberty and reproduction. Endline examinations additionally included four English language questions, and comprised materials taught in Secondary 1 and Secondary 2.

### 2.6. Study outcomes

Baseline menstrual-related absenteeism was assessed as self-reported number of school days missed due to menstruation, excluding days where the main reason for missing school was unrelated to menstruation ("Since schools reopened in January this year, how many days of class did you miss during your period?" "What is the main reason for missing

school during your period?"). The days of school missed for any reason was obtained from the question ("Since schools reopened in January this year, how many days of class did you miss for any reason?"). We calculated the number of days in the denominator from the official date schools re-opened post COVID-19 (10 January 2022) to the date of interview for each participant, excluding school holidays.

Endline menstrual-related absenteeism was assessed as self-reported number of school days missed during menstruation in Terms 1 and 2 of 2023 until the interview date. Term 1 was from 6 February to 5 May 2023, and Term 2 started on 29 May 2023. To minimize recall bias, we used cognitive testing before the survey to ensure clearly worded questions anchored to specific time frames, and provided reminders of school terms and holidays during survey completion. During self-completion of the questionnaires, research assistants were present to assist participants with clarifications about the recall period and absence reasons. Baseline and endline examination scores were calculated as the mean of subject-specific z-scores at each time point.

## 2.7. Statistical analyses

Data were exported to Stata version 18 for final cleaning and analysis. We estimated the mean number of days missed overall per month and the mean number of days reported to be missed due to menstruation using mixed-effects negative binomial regression models to estimate adjusted incidence rate ratios (aIRR) and 95% confidence intervals (CI), allowing for school-level clustering and over-dispersion. Similarly, we used mixed-effects linear regression models to estimate adjusted standardized mean differences (aSMD) and 95%CIs for examination score.

We built separate models for each group of exposure variables using a hierarchical conceptual framework based on the integrated model of menstrual experience [2] to guide adjustment for potential confounders (Fig 1). The dimensions of menstrual health are defined in Table 1. In line with this framework, we grouped variables into six hierarchical levels reflecting their conceptual proximity to the outcomes. We use the term "distal" to refer to variables hypothesized to causally precede and influence variables at more proximal levels: Level 1 (most distal – socio-demographic variables), Level 2 (social support during menstruation), Level 3 (menstrual knowledge, attitudes and adequate menstrual product use), Level 4 (menstrual pain management, menstrual practice needs, experience of menstrual-related teasing), Level 5 (Self-efficacy in addressing menstrual needs scale [SAMNS] [19] – menstrual care confidence with menstrual preparedness and pain management sub-scales) and Level 6 (most proximal – trouble concentrating in class) (Fig 1).

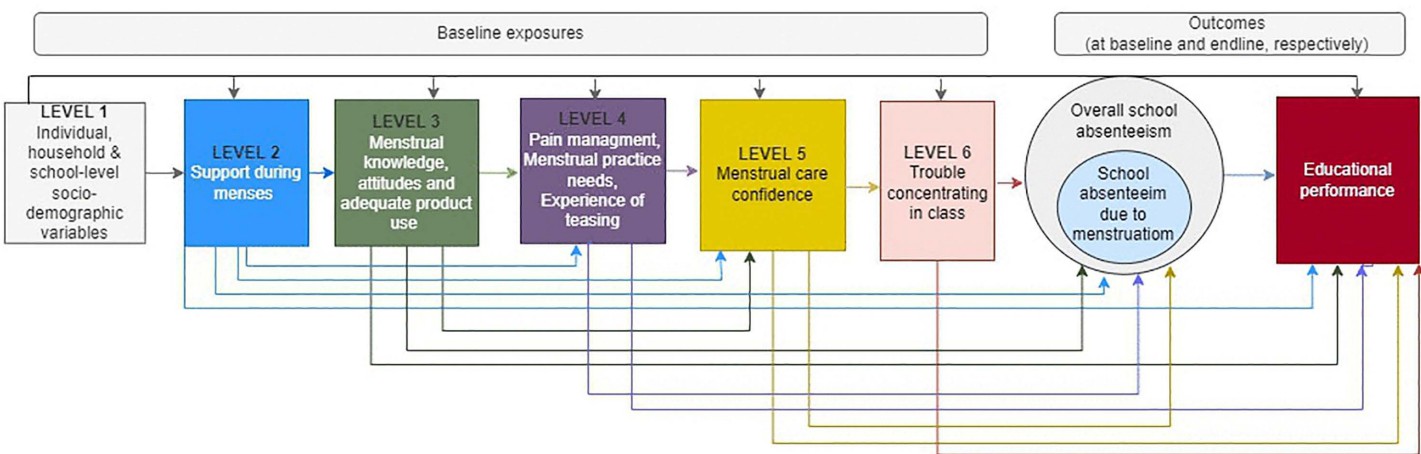

**Fig 1. Conceptual framework for statistical analysis.**

**Table 1. Definitions of menstrual-related exposure factors used in the conceptual framework.**

| Level | Survey measure | Brief description | Calculation | Range & interpretation |
|---|---|---|---|---|
| 2 | Social support for menstruation | An individual has someone who they feel okay asking for support for your period if needed (for advice, resources, emotional support) | Binary | Binary (yes/no) |
| 3 | Adequate product use | Used at least one adequate menstrual material and no inadequate materials at LMP | Binary | Binary (yes/no) |
| 3 | Knowledge of puberty & menstruation | Factual knowledge about puberty, menstruation, and the menstrual cycle | Number of 9 knowledge questions answered correctly | 0-9 (number of questions answered correctly) |
| 3 | Attitudes towards menstruation | Attitudes about what it is okay for people to do while menstruating and myths about painkillers | Number of 3 attitude questions answered positively | 0-3 (number of questions answered positively) |
| 4 | Pain at LMP | Reported having any pain at LMP. This includes reporting any pain, or a symptom of headache, backache, stomach ache or cramp at LMP. | Binary | Binary (yes/no) |
| 4 | Effective pain management strategy | Effective pain strategies defined as stretching, painkillers, eating foods with lots of water, exercising, drinking lots of clean water, holding a warm water bottle on the stomach. Ineffective methods defined as: doing nothing, taking antibiotics, eating spicy foods, drinking soda | Using at least one effective method and no ineffective methods, among girls reporting pain at LMP. | Binary (yes/no) |
| 4 | Menstrual Practice Needs Scale (MPNS-36) [21] | The extent to which individuals' menstrual management practices and environments were perceived to meet their needs during their LMP. | Weighted average of i) core items and school-specific items (75% weight) and ii) relevant material-specific items (25% weight). | Range 0–3. A higher score indicates fewer unmet menstrual needs (i.e., better than a lower score). |
| 4 | Pain relief | The amount of pain reduced | Binary | All/most or some/none |
| 4 | Experience of menstrual-related teasing | Whether boys or girls tease the participant about their period | Binary | Binary (yes/no) |
| 5 | Self-efficacy in addressing menstrual needs scale (SAMNS-26) [19] | Individuals' confidence in their capabilities to address their menstrual needs. This SAMNS scale has 3 sub-scales 1) The 17-item Menstrual Hygiene Preparation and Maintenance (MHPM) ("menstrual preparedness") sub-scale includes items regarding self-efficacy in preparing for menstruation (e.g., tracking one's cycle and anticipating days of bleeding), accomplishing various tasks related to maintaining menstrual hygiene (e.g., obtaining, using, cleaning, and disposing, menstrual materials in a variety of contexts), and seeking assistance when needed 2) the 4-item Executing Stigmatized Tasks (EST) sub-scale, which also included items related to obtaining and disposing menstrual materials and seeking assistance—but in situations where accomplishing the tasks involves greater risk of revealing one's menstrual status to a male person and 3) the 5-item Menstrual Pain Management (MPM), measured confidence in mitigating menstrual pain | Mean score | 0-100, representing the mean percentage confidence in addressing needs. Higher % is greater confidence |
| 6 | Trouble participating in class | Whether the participant had trouble participating in class due to period, during their most recent LMP at school | Binary | Binary (yes/no) |

We fitted multivariable, complete-case, models in which each variable was adjusted for other variables at the same hierarchical level and for variables in more distal (upstream) levels, as defined in the conceptual framework. An exception to this was the SAMNS sub-scales, which were not adjusted for each other. We used Wald tests to assess the strength of evidence and refer to a p-value<0.001 as indicating strong evidence of association [20]. We interpreted the results in conjunction with effect sizes and confidence intervals to provide comprehensive assessment of results.

To mitigate against the possibility of reverse causality, and to assess temporal associations of exposures with the outcomes, we analysed associations of baseline exposures with endline outcomes in the control arm, using analogous methods. We restricted these analyses to participants in the control arm as the MENISCUS intervention may have modified the association.

**2.7.1. Ethical approval.** Ethical approval for MENISCUS protocol, the informed consent forms and draft CRFs were obtained from the Uganda Virus Research Institute Research & Ethics Committee (UVRI-REC) (reference GC/127/819) and the Uganda National Council of Science and Technology (UNCST) (reference HS1525ES) and the LSHTM Research Ethics Committee (reference 22952−2).

**2.7.2. Consent for publication.** Participants consented to the research findings being published in international science journals and electronic websites, on condition that that findings cannot be traced to individuals.

## 3. Results

Of the 4281 eligible participants, 3878 (90.5%) consented/assented and completed the baseline survey (Fig 2). Of these, 3312 (79.1%) participants reported menstruating in the past 6 months and were included in the baseline analyses on school absenteeism. A further 367 (11.1%) participants were not present for the baseline UNEB exam and were excluded from baseline analyses of examination performance (Fig 2). Students who were missing baseline education performance data were more likely to have greater unmet menstrual needs (MPNS score, p=0.01) but there was no evidence of association with other dimensions of menstrual health.

### 3.1. Baseline characteristics of study population

The mean age of participants was 15.6 years (SD=0.9), with most (n=2161; 65.2%) attending a privately-owned school. Around half (n=2070; 55.3%) were day students and had their mother as the primary caregiver (n=1816; 54.8%). A total of 1227 (37.0%) participants lived in households of ≥8 members and 571 (17.2%) reported having one or fewer meals the previous day (Table 2). Overall, 2780 (83.9%) participants reported having social support during their last menstrual period (LMP) and 3038 (91.7%) reported exclusive use of adequate menstrual products at LMP (Table 3). A total of 1696 (51.2%) participants answered at least 2 of the 3 attitude questions positively, and 538 (16.2%) answered at least 7 of the 9 menstrual knowledge questions correctly.

Most participants (2446; 73.9%) reported experiencing menstrual pain at last menstrual period, of whom 1529 (62.5%) reported using an effective pain management strategy. The mean MPNS score of 2.11 (SD=0.52) corresponds to on average response of "sometimes" having unmet menstrual practice needs. The possible range is 0–3, with 3 corresponding to no unmet needs. Additionally, participants have a moderate level of menstrual care confidence regarding ability to manage menstrual needs (mean=60.8, SD=19.0). Overall, 251 (7.6%) and 284 (8.6%) participants reported menstrual-related teasing by boys and girls respectively, and almost half of participants reported trouble participating in class during periods (n=1414, 42.7%) (Table 3).

Participants reported missing a mean of 1.25 school days per month overall (95%CI 1.17, 1.33), and 0.30 days due to menstruation (95%CI 0.27, 0.34). A total of 991 (30%) participants reporting missing school due to menstruation since schools reopened, with 323 (9.8%) reporting missing at least one day per month due to menstruation.

### 3.2. Socio-demographic factors associated with menstrual-related absenteeism and examination performance at baseline

Menstrual-related absenteeism and poorer examination performance were both associated with being in a school with a poorer examination performance (aIRR=1.26, 95%CI 1.00, 1.60; aSMD=−0.44, 95%CI −0.57, −0.32), being a day student (aIRR=1.45, 95%CI 1.20, 1.77; aSMD=−0.12, 95%CI −0.18, −0.05), older age (aIRR-trend=1.20, 95%CI 1.09, 1.33; aSMD-trend=−0.10, 95%CI −0.13, −0.06) and having fewer meals the previous day (aIRR-trend=1.24, 95%CI 1.09,

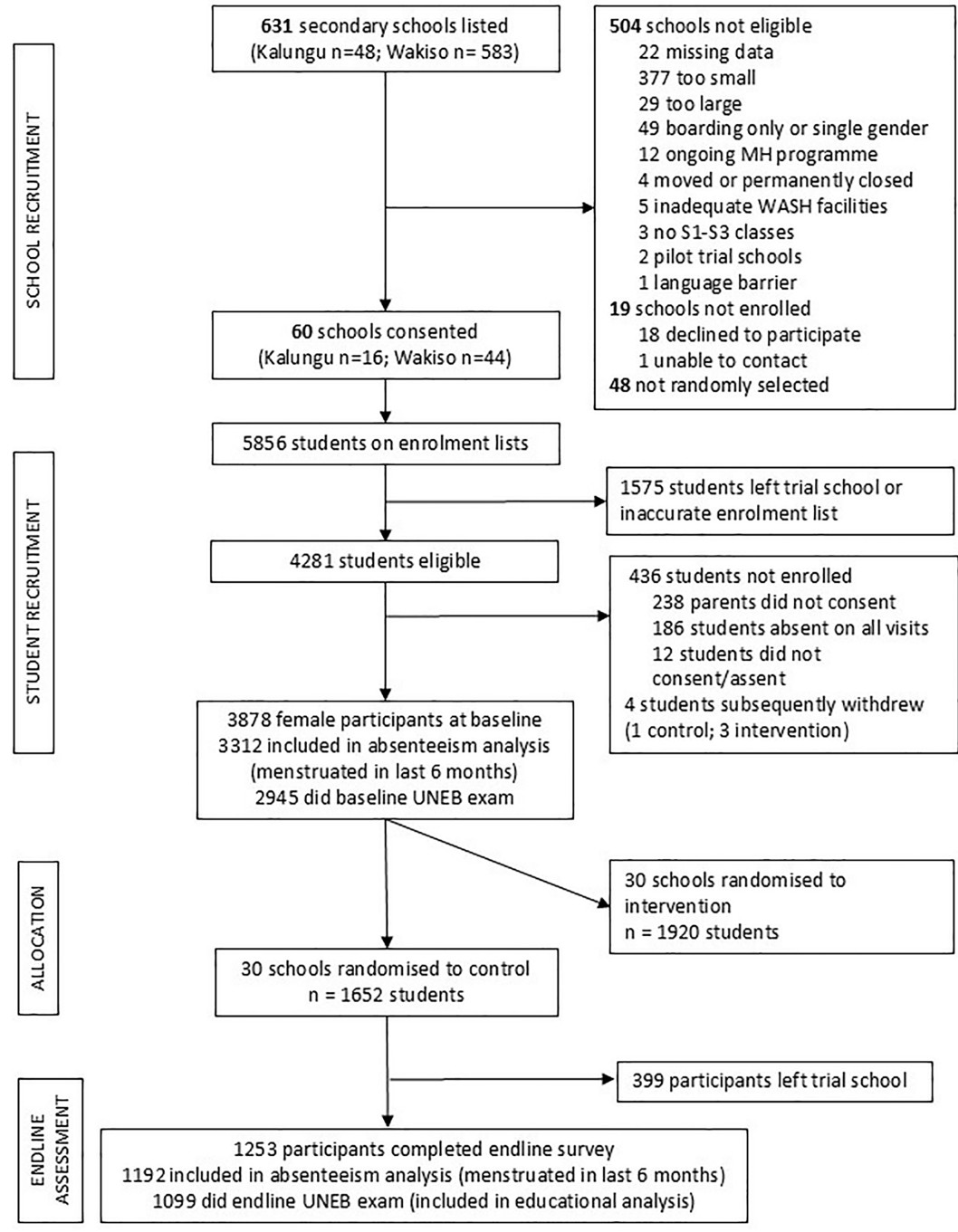

**Fig 2. Study flowchart.**

1.40; aSMD-trend = −0.06, 95%CI −0.10, −0.02) (Table 2). In addition, poorer examination performance was associated with older age at first menstruation (aSMD-trend = −0.05, 95%CI −0.08, −0.02) and higher education level of the primary caregiver (aSMD = −0.12, −0.19, −0.05) (Table 2).

**Table 2. Socio-demographic characteristics associated with school absenteeism due to menstruation and examination performance at baseline.**

| Variable | Number of participants | Days of school missed due to menstruation per month (n = 3312) | | Examination performance (n = 2945) | |
|---|---|---|---|---|---|
| | Frequency (%) | Mean day (SE) | aIRR (95%CI)[1] | Standardized mean exam score (SE) | aSMD (95%CI)[1] |
| **Structural-level characteristics** | | | | | |
| **District** | | | P = 0.38 | | P = 0.23 |
| Wakiso | 2600 (78.5) | 0.29 (0.02) | 1 | 0.03 (0.05) | 0 |
| Kalungu | 712 (21.5) | 0.37 (0.03) | 1.13 (0.86, 1.50) | −0.12 (0.10) | −0.09 (−0.23, 0.05) |
| **School-level characteristics** | | | | | |
| **School ownership** | | | P = 0.80 | | P = 0.99 |
| Private | 2161 (65.2) | 0.27 (0.02) | 1 | 0.07 (0.06) | 0 |
| Government | 1151 (34.8) | 0.38 (0.03) | 0.97 (0.74, 1.27) | −0.16 (0.05) | 0.00 (−0.14, 0.14) |
| **School-mean combined UNEB Math + Biology score** | | | P = 0.05 | | P < 0.001 |
| High UNEB | 1983 (53.0) | 0.25 (0.02) | 1 | 0.24 (0.05) | 0 |
| Low UNEB | 1758 (47.0) | 0.37 (0.02) | 1.26 (1.00, 1.60) | −0.24 (0.03) | −0.44 (−0.57, −0.32) |
| **Proportion of students boarding at school** | | | P = 0.95 | | P = 0.84 |
| < 50% | 1820 (55.0) | 0.35 (0.02) | 1 | −0.10 (0.06) | 0 |
| >=50% | 1492 (45.0) | 0.26 (0.02) | 0.99 (0.77, 1.28) | 0.12 (0.05) | 0.01 (−0.12, 0.15) |
| **School size (female baseline participants per school)** | | | P = 0.57 | | P = 0.24 |
| Greater or equal to 59 | 2258 (68.2) | 0.30 (0.02) | 1 | 0.03 (0.06) | −0.07 (−0.20, 0.05) |
| Fewer than 59 | 1054 (31.8) | 0.34 (0.02) | 0.93 (0.74, 1.18) | −0.04 (0.06) | 0 |
| **Student type** | | | P < 0.001 | | P = 0.001 |
| Boarding | 1496 (45.2) | 0.24 (0.02) | 1 | 0.11 (0.03) | 0 |
| Day | 1816 (54.8) | 0.37 (0.02) | 1.45 (1.20, 1.77) | −0.07 (0.05) | −0.12 (−0.18, −0.05) |
| **Individual-level socio-demographic & economic factors** | | | | | |
| **Age group** | | | P-trend = 0.004 | | P-trend = 0.003 |
| <15 | 283 (8.5) | 0.22 (0.03) | 1 | 0.19 (0.07) | 0.00 |
| 15 | 1353 (40.9) | 0.25 (0.02) | 1.15 (0.83, 1.60) | 0.11 (0.05) | −0.02 (−0.13, 0.08) |
| 16 | 1234 (37.3) | 0.36 (0.02) | 1.54 (1.10, 2.15) | −0.08 (0.04) | −0.17 (−0.28, −0.06) |
| 17 | 341 (10.3) | 0.38 (0.05) | 1.66 (1.11, 2.49) | −0.14 (0.06) | −0.23 (−0.36, −0.09) |
| 18+ | 101 (3.0) | 0.48 (0.12) | 1.71 (0.97, 3.03) | −0.35 (0.10) | −0.34 (−0.53, −0.15) |
| Trend | | | 1.20 (1.09, 1.33) | | −0.10 (−0.13, −0.06) |
| **Age at first menstruation** | | | P = 0.02 | | P-trend = 0.003 |
| <13 years | 758 (22.9) | 0.34 (0.03) | 1 | 0.09 (0.06) | 0.00 |
| 13 years | 1230 (37.1) | 0.24 (0.02) | 0.74 (0.59, 0.92) | 0.05 (0.05) | −0.01 (−0.08, 0.06) |
| 14 years | 1027 (31.0) | 0.34 (0.03) | 0.92 (0.73, 1.16) | −0.07 (0.05) | −0.08 (−0.16, −0.00) |
| >14 years | 297 (9.0) | 0.40 (0.05) | 1.02 (0.73, 1.42) | −0.20 (0.06) | −0.14 (−0.25, −0.03) |
| Trend | | | n/a | | −0.05 (−0.08, −0.02) |
| **Religion** | | | P = 0.002 | | P = 0.19 |
| Catholic | 1052 (31.8) | 0.28 (0.02) | 1 | −0.03 (0.05) | 0 |
| Protestant/SDA | 1285 (38.8) | 0.29 (0.02) | 1.09 (0.88, 1.35) | 0.03 (0.05) | 0.06 (−0.01, 0.13) |
| Muslim | 959 (29.0) | 0.36 (0.03) | 1.45 (1.14, 1.84) | −0.05 (0.05) | −0.01 (−0.09, 0.06) |
| None/Other | 16 (0.5) | 0.87 (0.44) | 3.78 (1.22, 11.74) | 0.03 (0.27) | 0.07 (−0.38, 0.53) |
| **Ethnicity** | | | P = 0.54 | | P = 0.83 |
| Muganda | 2280 (68.8) | 0.32 (0.02) | 1 | 0.00 (0.05) | 0 |
| Non Muganda | 1032 (31.2) | 0.30 (0.02) | 0.94 (0.79, 1.14) | 0.02 (0.04) | 0.01 (−0.05, 0.07) |

*(Continued)*

**Table 2.** (Continued)

| Variable | Number of participants | Days of school missed due to menstruation per month (n=3312) | | Examination performance (n=2945) | |
|---|---|---|---|---|---|
| | Frequency (%) | Mean day (SE) | aIRR (95%CI)[1] | Standardized mean exam score (SE) | aSMD (95%CI)[1] |
| **Primary caregiver** | | | P=0.17 | | P=0.22 |
| Mother | 1938 (58.5) | 0.33 (0.02) | 1 | 0.00 (0.05) | 0 |
| Father | 805 (24.3) | 0.27 (0.02) | 0.85 (0.69, 1.05) | 0.06 (0.05) | 0.04 (−0.02, 0.11) |
| Self and others | 569 (17.2) | 0.29 (0.03) | 0.84 (0.67, 1.06) | −0.10 (0.05) | −0.03 (−0.11, 0.05) |
| **Education of primary caregiver** | | | P=0.15 | | P=0.001 |
| Secondary or more | 2215 (59.2) | 0.29 (0.02) | 1 | −0.03 (0.05) | 0 |
| Primary or less | 881 (23.6) | 0.38 (0.03) | 1.09 (0.88, 1.34) | 0.04 (0.06) | 0.12 (0.05, 0.19) |
| Don't know | 645 (17.2) | 0.28 (0.03) | 0.83 (0.66, 1.05) | 0.05 (0.05) | 0.09 (0.02, 0.17) |
| **Household size** | | | P=0.19 | | P=0.89 |
| 0-5 | 1001 (30.2) | 0.28 (0.02) | 1 | −0.007 (0.05) | −0.01 (−0.08, 0.05) |
| 6-7 | 1084 (32.7) | 0.31 (0.03) | 1.20 (0.97, 1.49) | −0.001 (0.05) | −0.01 (−0.08, 0.05) |
| >=8 | 1227 (37.0) | 0.34 (0.03) | 1.18 (0.95, 1.45) | 0.00 (0.04) | 0 |
| **Number of meals eaten on the previous day** | | | P-trend=0.008 | | P-trend=0.005 |
| Three or more | 1058 (31.9) | 0.24 (0.02) | 1 | 0.05 (0.05) | 0 |
| Two | 1683 (50.8) | 0.31 (0.02) | 1.20 (0.99, 1.46) | 0.001 (0.05) | −0.03 (−0.09, 0.04) |
| One or fewer | 571 (17.2) | 0.41 (0.04) | 1.53 (1.19, 1.97) | −0.08 (0.05) | −0.13 (−0.21, −0.04) |
| Trend | | | 1.24 (1.09, 1.40) | | −0.06 (−0.10, −0.02) |
| **Socioeconomic status quintile** | | | P=0.02 | | P=0.04 |
| Highest | 656 (19.8) | 0.24 (0.03) | 1 | 0.005 (0.06) | 0 |
| Medium-high | 684 (20.7) | 0.25 (0.03) | 0.91 (0.69, 1.19) | 0.14 (0.04) | 0.12 (0.04, 0.21) |
| Medium | 655 (19.8) | 0.29 (0.03) | 0.99 (0.75, 1.30) | 0.009 (0.06) | 0.04 (−0.05, 0.13) |
| Low-medium | 676 (20.4) | 0.30 (0.03) | 0.92 (0.69, 1.22) | −0.02 (0.05) | 0.03 (−0.06, 0.13) |
| Low | 641 (19.4) | 0.47 (0.04) | 1.35 (1.01, 1.81) | −0.06 (0.05) | 0.00 (−0.09, 0.10) |

[1]Adjusted for all socio-demographic factors and school level clustering.

### 3.3. Menstrual factors associated with school absenteeism due to menstruation and poorer examination performance at baseline

Among the 991 participants reporting missing school due to menstruation, 1023 days were missed per month (mean 1.03 per month per participant (SE=0.04) (Table 4). The most commonly-reported main reason was illness (back/stomach pain or cramps: n=588; 57.2% of days missed; feeling generally unwell: n=170; 17.2% of days missed). Relatively few participants reported other reasons (stigma-related, the physical environment or socio-economic) as the main reason for missing school during menstruation (Table 4).

After adjusting for variables at the same hierarchical levels of the hierarchical framework, there was strong evidence that menstrual-related absenteeism and poorer examination performance were both associated with not exclusively using adequate menstrual products (aIRR=1.82, 95%CI 1.37, 2.42; aSMD=−0.13, 95%CI −0.23, −0.03), negative menstrual attitudes (aIRR=1.42, 95%CI 1.21, 1.68, aSMD=−0.15, 95%CI −0.21, −0.10), menstrual pain at LMP (aIRR=2.12, 95%CI 1.73, 2.59; aSMD=0.06, 95%CI 0.00, 0.13), more unmet menstrual practice needs (p-trend<0.001) and reported experience of menstrual-related teasing at LMP by girls (aIRR=1.31, 95%CI 0.98, 1.75; aSMD=−0.17, 95%CI −0.27, −0.07) (Table 3).

**Table 3. Menstrual characteristics associated with school absenteeism due to menstruation and examination performance at baseline.**

| Variable | Number of participants | Days of school missed due to menstruation per month (n = 3312) | | | Examination performance (n = 2945) | |
|---|---|---|---|---|---|---|
| | Frequency (%) | Mean day (SE) | aIRR (95%CI)[1] | | Standardized mean exam score (SE) | aSMD (95%CI)[1] |
| **Level 2 – Menstrual support** | | | | | | |
| **Social support for menstruation** | | | P = 0.77 | | | P = 0.28 |
| Yes | 2780 (83.9) | 0.30 (0.01) | 1 | | −0.007 (0.04) | 0 |
| No | 532 (16.1) | 0.35 (0.04) | 1.03 (0.82, 1.31) | | 0.03 (0.06) | 0.04 (−0.03, 0.12) |
| **Level 3 – Menstrual knowledge, attitudes and adequate product use** | | | | | | |
| **Adequate product use at LMP** | | | P < 0.001 | | | P = 0.008 |
| Yes | 3038 (91.7) | 0.29 (0.02) | 1 | | 0.009 (0.04) | 0 |
| No | 274 (8.3) | 0.54 (0.07) | 1.82 (1.37, 2.42) | | −0.14 (0.06) | −0.13 (−0.23, −0.03) |
| **Attitudes towards menstruation** | | | P < 0.001 | | | P < 0.001 |
| Positive (2–3) | 1696 (51.2) | 0.25 (0.02) | 1 | | 0.10 (0.05) | 0 |
| Negative (0–1) | 1616 (48.8) | 0.37 (0.02) | 1.42 (1.21, 1.68) | | −0.10 (0.05) | −0.15 (−0.21, −0.10) |
| **Knowledge of puberty and menstruation** | | | P-trend = 0.58 | | | P-trend<0.001 |
| High (7–9) | 538 (16.2) | 0.27 (0.04) | 1 | | 0.34 (0.07) | 0 |
| Medium (4–6) | 2378 (71.8) | 0.31 (0.02) | 1.05 (0.83, 1.33) | | −0.03 (0.04) | −0.32 (−0.40, −0.25) |
| Low (0–3) | 396 (12.0) | 0.39 (0.05) | 1.09 (0.79, 1.51) | | −0.30 (0.05) | −0.55 (−0.66, −0.45) |
| Trend | | | 1.06 (0.86, 1.30) | | | −0.28 (−0.33, −0.23) |
| **Level 4 – Menstrual pain, pain management and experience of menstrual-related teasing** | | | | | | |
| **Menstrual pain at LMP** | | | p < 0.001 | | | P = 0.04 |
| No | 866 (26.1) | 0.15 (0.02) | 1 | | 0.02 (0.05) | 0 |
| Yes | 2446 (73.9) | 0.37 (0.02) | 2.12 (1.73-2.59) | | 0.001 (0.05) | 0.06 (0.00, 0.13) |
| **Effective pain management strategy at LMP** | | | P < 0.001 | | | P = 0.09 |
| No pain | 866 (26.1) | 0.15 (0.02) | 1 | | 0.02 (0.05) | 0 |
| Pain and used at least one effective management strategy | 1529 (46.2) | 0.35 (0.02) | 2.13 (1.72, 2.63) | | −0.02 (0.04) | 0.08 (0.01, 0.14) |
| Pain and did not use at least one effective strategy | 917 (27.7) | 0.39 (0.03) | 2.10 (1.66, 2.66) | | −0.06 (0.04) | 0.05 (−0.03, 0.12) |
| **Menstrual pain relief** | | | P < 0.001 | | | P = 0.13 |
| No pain | 866 (26.1) | 0.15 (0.02) | 1 | | 0.02 (0.05) | 0 |
| All/most pain relieved | 983 (29.7) | 0.34 (0.03) | 2.00 (1.59, 2.51) | | 0.01 (0.05) | 0.06 (−0.01, 0.13) |
| Some/none of pain relieved | 1463 (44.2) | 0.38 (0.02) | 2.20 (1.78, 2.73) | | −0.005 (0.04) | 0.07 (−0.00, 0.14) |
| **Menstrual Practice Needs Score (MPNS score)** | | | P-trend<0.001 | | | P-trend<0.001 |
| High – few unmet needs (2.38–3.00] | 1116 (34.0) | 0.18 (0.02) | 1 | | 0.12 (0.05) | 0 |
| Medium (1.88–2.38] | 1091 (33.2) | 0.27 (0.02) | 1.41 (1.15, 1.73) | | −0.03 (0.04) | −0.10 (−0.17, −0.04) |
| Low – many unmet needs (0.00–1.88] | 1076 (32.8) | 0.48 (0.03) | 2.02 (1.63, 2.50) | | −0.09 (0.05) | −0.14 (−0.21, −0.07) |
| Trend | | | 1.44 (1.29-1.57) | | | −0.07 (−0.11, −0.03) |
| **Experience of menstrual-related teasing by boys** | | | P = 0.04 | | | P = 0.18 |
| No | 3061 (92.4) | 0.29 (0.01) | 1 | | 0.01 (0.04) | 0 |
| Yes | 251 (7.6) | 0.60 (0.08) | 1.38 (1.02, 1.88) | | −0.20 (0.05) | −0.07 (−0.18, 0.03) |
| **Experience of menstrual-related teasing by girls** | | | P = 0.07 | | | p < 0.001 |
| No | 3028 (91.4) | 0.29 (0.01) | 1 | | 0.02 (0.04) | 0 |
| Yes | 284 (8.6) | 0.53 (0.07) | 1.31 (0.98,1.75) | | −0.30 (0.06) | −0.17 (−0.27, −0.07) |

*(Continued)*

**Table 3.** (Continued)

| Variable | Number of participants | Days of school missed due to menstruation per month (n = 3312) | | | Examination performance (n = 2945) | |
|---|---|---|---|---|---|---|
| | Frequency (%) | Mean day (SE) | aIRR (95%CI)[1] | | Standardized mean exam score (SE) | aSMD (95%CI)[1] |
| **Level 5 – Menstrual care confidence** | | | | | | |
| **Menstrual care confidence (SAMNS score)** | | | P = 0.01 | | | P = 0.73 |
| High self-efficacy (69.62–100] | 1129 (34.1) | 0.25 (0.02) | 1 | | 0.05 (0.05) | 0 |
| Medium (52.31–69.62] | 1102 (33.3) | 0.25 (0.02) | 0.88 (0.72, 1.08) | | −0.02 (0.05) | −0.01 (−0.07, 0.06) |
| Low self-efficacy (0–52.31] | 1081 (32.6) | 0.43 (0.03) | 1.20 (0.98, 1.49) | | −0.03 (0.05) | 0.02 (−0.05, 0.09) |
| **Menstrual preparedness sub-scale** | | | P-trend<0.001 | | | P = 0.86 |
| High (76.47–100] | 1058 (31.9) | 0.21 (0.02) | 1 | | 0.07 (0.05) | 0 |
| Medium (58.25–76.47]- | 1127 (34.0) | 0.26 (0.02) | 1.15 (0.93, 1.41) | | 0.002 (0.05) | 0.01(−0.05, 0.08) |
| Low (0–58.25] | 1127 (34.0) | 0.45 (0.03) | 1.54 (1.24, 1.91) | | −0.06 (0.05) | 0.00 (−0.08, 0.07) |
| Trend | | | 1.25 (1.12, 1.39) | | | n/a |
| **Pain management sub-scale** | | | P = 0.49 | | | P = 0.71 |
| High (82–100] | 923 (27.9) | 0.28 (0.03) | 1 | | −0.001 (0.05) | 0 |
| Medium (50–82] | 1241 (37.5) | 0.28 (0.02) | 0.98 (0.80, 1.21) | | 0.02 (0.05) | 0.02 (−0.04, 0.09) |
| Low (0–50] | 1148 (34.7) | 0.37 (0.03) | 1.10 (0.89, 1.36) | | −0.02 (0.05) | 0.03 (−0.04, 0.10) |
| **Level 6 – Class participation and attendance at LMP** | | | | | | |
| **Trouble participating in class during LMP** | | | P<0.001 | | | P = 0.91 |
| No | 1898 (57.3%) | 0.20 (0.01) | 1 | | 0.007 (0.04) | 0 |
| Yes | 1414 (42.7%) | 0.46 (0.03) | 1.80 (1.51, 2.14) | | −0.01 (0.05) | 0.00 (−0.06, 0.06) |
| **School absenteeism due to menstruation** | | | | | | P = 0.01 |
| Missed less than one day per month | n/a | n/a | n/a | | 0.14 (0.04) | 0 |
| Missed at least one day per month | n/a | n/a | n/a | | −0.20 (0.05) | −0.12 (−0.22, −0.03) |

[1]Adjusted for school level clustering, level 1 (socio-demographic) factors and variables at the same or more distal levels. SAMNS scores and sub-scales are not adjusted for each other. Trouble participating, and school absenteeism are not adjusted for each other.

Menstrual-related absenteeism was also associated with lack of perceived pain relief at LMP (aIRR = 2.20, 95%CI 1.78, 2.73 for minimal/no pain relieved vs no pain), menstrual-related teasing by boys (aIRR = 1.38, 95%CI 1.02, 1.88), poor menstrual care confidence (aIRR = 1.20, 95%CI 0.98, 1.49 for lowest vs highest quartile, p-trend = 0.01), especially due to lack of menstrual preparedness (aIRR = 1.54, 95%CI 1.24, 1.91, p-trend<0.001) and with trouble participating in class during LMP (aIRR = 1.80, 95%CI 1.51, 2.14). In absolute terms, the differences were relatively small, for example, the mean days missed per month among participants with no menstrual pain at LMP was 0.15 (SE 0.02), compared with 0.37 (SE 0.02) among participants with pain at LMP. Similar factors were associated with missing school overall (i.e., for any reason) (S1 Table).

Poorer examination performance was also associated with school absenteeism (aSMD = −0.12, 95%CI −0.22, −0.03 for ≥1 vs < 1 day missed per month due to menstruation).

### 3.4. Menstrual factors at baseline associated with menstrual-related absenteeism and poorer examination performance at endline

Among the 1652 participants in control arm schools at baseline, 1253 (75.8%) were in control schools at endline and completed the endline survey (Fig 2). Of these, 1192 (95.1%) reported having menstruated in the past 6 months and

**Table 4. Main reason for missing school due to menstruation, reported at baseline.**

| Main reason for missing school due to menstruation | Number of participants n (%) | Mean (SD) days per participant missed due to menstruation per month among those reporting this reason | Total days missed due to menstruation per month with this as the main reason | Proportion of days missed due to menstruation per month with this as the mains reason |
|---|---|---|---|---|
| **Total** | **991** | **1.03 (1.16)** | **1023** | **100%** |
| **Illness** | | | | |
| Back/stomach pain or cramps | 588 (59.3) | 1.00 (1.06) | 585 | 57.2% |
| Feeling generally unwell | 170 (17.2) | 0.98 (1.20) | 165 | 16.2% |
| **Stigma** | | | | |
| Scared of leaking blood on clothes | 59 (6.0) | 0.92 (0.95) | 54 | 5.3% |
| Not wanting to change pad or absorbents at school | 2 (0.2) | 2.49 (3.01) | 5 | 0.5% |
| Household members telling me not to go to school during period | 12 (1.2) | 1.86 (2.38) | 22 | 2.2% |
| Worried that others may know I have my periods | 18 (1.8) | 0.79 (0.53) | 14 | 1.4% |
| Scared of having to ask for help because of periods | 6 (0.6) | 0.94 (0.72) | 6 | 0.5% |
| **Physical environment** | | | | |
| School toilets or latrines not clean | 34 (3.4) | 1.20 (1.28) | 41 | 4.0% |
| Lack of access to toilet paper at school | 4 (0.4) | 0.56 (0.24) | 2 | 0.2% |
| Lack of access to water and/or soap at school | 17 (1.7) | 1.34 (2.27) | 23 | 2.2% |
| Not enough privacy in school toilet/latrine | 34 (1.4) | 1.20 (1.28) | 41 | 4.0% |
| **Socio-economic** | | | | |
| Unable to afford pads or other absorbents | 35 | 1.50 (1.45) | 52 | 5.1% |
| Lack of knickers | 31 | 1.26 (1.35) | 39 | 4.8% |

were included in endline analysis, and 1099 (92.2%) had examination data performance at endline (Fig 2). The median follow-up duration was 394 days (interquartile range 381–441 days).

Among these participants, 135 (11.3%) reported missing at least one day of school due to menstruation, and the mean days missed was 0.31 per month (95%CI 0.27–0.37). Menstrual-related absenteeism and poorer examination performance at end-line were both associated with using inadequate menstrual products at baseline (aIRR=1.76, 95%CI 0.95, 3.24; aSMD=−0.19, 95%CI −0.34, −0.03), negative menstrual attitudes (aIRR=1.76, 95%CI 1.26, 2.46; aSMD=−0.16, 95%CI −0.24, −0.08), more unmet menstrual practice needs (aIRR=1.75, 95%CI 1.12–2.71; aSMD=−0.10, 95%CI −0.21 to 0.00) and experience of menstrual-related teasing by girls (aIRR=1.89, 95%CI 1.06, 2.73; aSMD=−0.33, 95%CI −0.48, −0.18) (Table 5).

In addition, menstrual-related absenteeism was associated with baseline menstrual pain (aIRR=3.08, 95%CI 2.04, 4.65), including if there was pain management but no pain relief (aIRR=3.42, 95%CI 2.18, 5.36 vs no pain), and trouble participat-ing in class during LMP (aIRR=1.69, 95%CI 1.20, 2.38). Endline examination performance was additionally associated with poorer baseline menstrual knowledge (aSMD=−0.44, 95%CI −0.60, −0.28) (Table 5). Similar factors were associated with school absenteeism for any reason (including menstruation) (S1 Table). For example, the mean number of days missed due to menstrual pain was 1.36 days per month, vs 0.88 among those without pain at LMP (aIRR=1.41, 95%CI 1.26–1.57).

## 4. Discussion

### 4.1. Multiple dimensions of menstrual health are associated with school absenteeism and poor examination performance

In this setting, there is strong evidence that use of inadequate menstrual products, negative attitudes towards menstrua-tion, having many unmet menstrual needs, menstrual pain and experience of menstrual-related teasing were associated

**Table 5. Baseline menstrual characteristics associated with endline school absenteeism due to menstruation and examination performance.**

| Variable | Number of participants | Days of school missed due to menstruation at endline (n=1192) | | Examination performance at endline (n=1099) | |
|---|---|---|---|---|---|
| | | Frequency (%) | Mean day (SE) | aIRR (95%CI)[1] | Standardized mean exam score (SE) | aSMD (95%CI)[1] |
| **Level 2 – Menstrual support** | | | | | | |
| **Social support for menstruation** | | | | P=0.18 | | P=0.15 |
| Yes | 1010 (84.7) | 0.32 (0.03) | 1 | −0.05 (0.08) | 0 |
| No | 182 (15.3) | 0.25 (0.04) | 0.73 (0.45, 1.16) | −0.06 (0.09) | −0.09 (−0.03, 0.20) |
| **Level 3 – Menstrual knowledge, attitudes and adequate product use** | | | | | | |
| **Adequate product use** | | | | P=0.07 | | P=0.01 |
| Yes | 1106 (92.8) | 0.30 (0.02) | 1 | −0.08 (0.08) | 0 |
| No | 86 (7.2) | 0.48 (0.12) | 1.76 (0.95, 3.24) | −0.23 (0.11) | −0.19 (−0.34, −0.03) |
| **Attitudes towards menstruation** | | | | P<0.001 | | P<0.001 |
| Positive (2–3) | 614 (51.5) | 0.23 (0.03) | 1 | 0.01 (0.08) | 0 |
| Negative (0–1) | 578 (48.5) | 0.40 (0.04) | 1.76 (1.26, 2.46) | −0.18 (0.08) | −0.16 (−0.24, −0.08) |
| **Knowledge of puberty and menstruation** | | | | P=0.72 | | P-trend<0.001 |
| High (7–9) | 198 (16.6) | 0.26 (0.05) | 1 | 0.26 (0.09) | 0 |
| Medium (4–6) | 839 (70.4) | 0.32 (0.03) | 1.13 (0.72, 1.79) | −0.06 (0.07) | −0.22 (−0.33, −0.10) |
| Low (0–3) | 155 (13.0) | 0.36 (0.08) | 0.96 (0.50, 1.83) | −0.29 (0.09) | −0.44 (−0.60, −0.28) |
| Trend | | | | | | −0.22 (−0.30, −0.14) |
| **Level 4 – Menstrual pain, pain management and experience of teasing** | | | | | | |
| **Menstrual pain at LMP** | | | | P<0.001 | | P=0.68 |
| No pain | 305 (25.6) | 0.13 (0.04) | 1 | 0.03 (0.09) | 0 |
| Any pain | 887 (74.4) | 0.37 (0.03) | 3.08 (2.04, 4.65) | −0.07 (00.22, 0.09) | −0.01 (−0.11, 0.08) |
| **Effective pain management strategy** | | | | P<0.001 | | P=0.93 |
| No pain | 305 (25.6) | 0.13 (0.04) | 1 | 0.03 (0.09) | 0 |
| Pain and used at least one effective management strategy | 538 (45.1) | 0.39 (0.04) | 3.07 (1.98, 4.76) | −0.06 (0.09) | −0.01 (−0.11, 0.10) |
| Pain and did not use at least one effective strategy | 349 (29.3) | 0.36 (0.05) | 3.09 (0.95, 2.15) | −0.07 (0.08) | −0.02 (−0.13, 0.09) |
| **Menstrual pain relief** | | | | P<0.001 | | P=0.64 |
| No pain | 305 (25.6) | 0.13 (0.04) | 1 | 0.03 (0.09) | 0 |
| All/most pain relieved | 371 (31.1) | 0.35 (0.05) | 2.72 (1.72, 4.29) | −0.11 (0.10) | −0.04 (−0.15, 0.07) |
| Some/none of pain relieved | 516 (43.3) | 0.39 (0.03) | 3.42 (2.18, 5.36) | −0.04 (0.08) | 0.01 (−0.10, 0.11) |
| **Menstrual Practice Needs Score (MPNS score)** | | | | P=0.005 | | P=0.06 |
| High – few unmet needs (2.38–3.00] | 445 (37.6) | 0.20 (0.03) | 1 | 0.05 (0.08) | 0 |
| Medium (1.88–2.38] | 372 (31.4) | 0.35 (0.05) | 1.43 (0.95, 2.15) | −0.07 (0.09) | −0.07 (−0.17, 0.04) |
| Low – many unmet needs (0.00–1.88] | 366 (30.9) | 0.42 (0.04) | 1.75 (1.12, 2.71) | −0.13 (0.08) | −0.10 (−0.21, 0.00) |
| **Experience of menstrual-related teasing by boys** | | | | P=0.18 | | P=0.25 |
| No | 1098 (92.1) | 0.29 (0.02) | 1 | −0.04 (0.08) | 0 |
| Yes | 94 (7.9) | 0.59 (0.12) | 1.50 (0.83, 2.73) | −0.16 (0.13) | 0.07 (−0.09, 0.22) |
| **Experience of menstrual-related teasing by girls** | | | | P=0.03 | | P<0.001 |
| No | 1087 (91.2) | 0.29 (0.02) | 1 | −0.02 (0.08) | 0 |
| Yes | 105 (8.8) | 0.56 (0.09) | 1.89 (1.06, 3.37) | −0.35 (0.12) | −0.33 (−0.48, −0.18) |

*(Continued)*

**Table 5.** (Continued)

| Variable | Number of participants | Days of school missed due to menstruation at endline (n = 1192) | | Examination performance at endline (n = 1099) | |
|---|---|---|---|---|---|
| | Frequency (%) | Mean day (SE) | aIRR (95%CI)[1] | Standardized mean exam score (SE) | aSMD (95%CI)[1] |
| **Level 5 – Menstrual care confidence** | | | | | |
| **Menstrual care confidence (SAMNS score)** | | | p-trend = 0.40 | | P = 0.55 |
| High self-efficacy (69.62–100) | 420 (35.2) | 0.28 (0.04) | 1 | 0.02 (0.09) | 0 |
| Medium (52.31–69.62) | 397 (33.3) | 0.30 (0.04) | 1.08 (0.71, 1.62) | −0.02 (0.09) | 0.02 (−0.08, 0.12) |
| Low (0–52.31) | 375 (31.5) | 0.36 (0.04) | 1.20 (0.78, 1.83) | −0.13 (0.08) | −0.04 (−0.15, 0.07) |
| Trend | | | 0.91 (0.74, 1.13) | | |
| **Level 6 – Class participation and attendance at LMP** | | | | | |
| **Trouble participating in class during LMP** | | | P < 0.001 | | P = 0.52 |
| No | 678 (56.9) | 0.20 (0.02) | 1 | −0.05 (0.08) | 0 |
| Yes | 514 (43.1) | 0.47 (0.04) | 1.69 (1.20-2.38) | −0.06 (0.08) | 0.03 (−0.06, 0.12) |
| **School absenteeism due to menstruation** | n/a | n/a | n/a | | P = 0.43 |
| Missed less than one day per month | n/a | n/a | n/a | −0.04 (0.08) | 0 |
| Missed at least one day per month | n/a | n/a | n/a | −0.20 (0.10) | −0.06 (−0.22, 0.09) |

[1]Adjusted for school level clustering and baseline variables at Level 1 (socio-demographic) and at the same or more distal levels.

with both increased menstrual-related absenteeism and poorer examination performance. These results add to the limited evidence quantifying both the amount of menstrual-related absenteeism, and the specific dimensions of menstrual health associated with absenteeism and examination performance.

We have previously shown that a multi-component menstrual health intervention improved dimensions of menstrual health including knowledge, attitudes, self-efficacy and pain management, there was no evidence of an intervention effect on educational outcomes (absenteeism and examination performance) [14]. Through this observational analysis, we have shown that multiple dimensions of menstrual health are associated with educational outcomes, but the absolute effect was small, which may help explain why the intervention was not sufficient to improve these educational outcomes.

### 4.2. The impact of menstrual health on number of days of school missed due to menstruation

Our finding that 10% of participants reported missing at least one school day per month due to menstruation supports the widespread claim that one in 10 girls miss school due to menstruation [3]. However, on average, participants reported missing less than half a day of school per month due to menstruation (0.30 days). The prevalence of menstrual-related absenteeism is at the lower end of the range seen in the systematic review of studies in Sub-Saharan Africa (6–65% missing school during menstruation) [5], and slightly lower than the recent multi-country study (17.7%, 95%CI 15.1–20.3%) among 15–19 year olds [6]. Reported school absence may be lower in our study because i) students had missed almost two years of school due to COVID-19 closures and may have felt pressure from teachers and parents to make up the time; ii) reported days of school missed due to menstruation may be under-reported by boarding students who interpreted being on school premises as attending school, and iii) the relatively long period of reporting in our study may have led to underestimation due to recall bias. The variation in estimates between studies is also likely due to challenges measuring school attendance accurately [8]. Measuring school attendance using self-report often underestimates absenteeism compared to using school registers [22], which may be due to social desirability bias [8,23], stigma and taboos around menstruation, and recall bias. This is supported by qualitative findings from a Kenyan study, in which girls rarely reported

school absenteeism themselves, but reported that other girls were absent from school due to menstruation [24]. However in some settings including Uganda, school registers are incomplete and/or inaccurate, due to teachers not completing the registers, and social desirability bias if school funding is linked with student numbers [22]. Future studies should consider a combination of these methods, daily diaries and spot checks to triangulate data on school attendance [22].

To our knowledge, this is the first study in sub-Saharan Africa to quantify days of school absenteeism attributable to specific dimensions of menstrual health. We found that students reporting menstrual pain had more than twice the rate of menstrual-related absenteeism than those with no pain, and school absenteeism was highest among those without adequate pain relief. Further, we found that menstrual pain or feeling generally unwell were responsible for about three-quarters of days of school absenteeism due to menstruation. At baseline, there was evidence that participants with low menstrual self-efficacy were more likely to miss school due menstruation compared to those with high menstrual self-efficacy. The association was strongest with the menstrual preparedness sub-scale on the menstrual self-efficacy scale (Table 1). This fits with findings from qualitative studies that students with low menstrual confidence might feel uncomfortable to be around students or teachers during menstruation and hence choose to miss school during menstruation [9,25]. This association may also be due to girls fearing leakage of blood whilst in class and hence missing school [2].

### 4.3. The impact of menstrual health on examination performance

Qualitative studies have described how participants perceive that menstruation affects their educational outcomes [2] but there are few rigorous quantitative studies on this. Our findings broadly support the hypothesis that menstrual health affects educational outcomes through multiple pathways, through menstrual pain or use of inadequate materials, but also due to social stigma and lack of menstrual knowledge, and through school absenteeism. Using a standardised assessment set by the UNEB, we found that poorer examination performance was associated with use of inadequate products, negative attitudes towards menstruation, having many unmet menstrual needs, menstrual pain and experience of menstrual-related teasing. There was strong evidence of associations between poorer examination scores and use of inadequate materials, poorer menstrual attitudes, poorer menstrual knowledge, having more unmet menstrual practice needs and experience of being teased in relation to periods. However, overall, associations of dimensions of menstrual health with examination scores were weaker than with school absenteeism. This is as expected given that absenteeism is more proximal to poor menstrual health than educational performance.

Our results support qualitative findings from the systematic review [2] and a qualitative study of 120 girls aged 14–16 years in in Kenya where girls reported that their attention in class was disturbed by worries of leakage of menstrual blood and staining of clothing [24]. A further study found that students reported failing to stand up and answer questions, write on the board and miss exams during menstruation [26]. Missing at least one day of school due to menstruation at LMP at baseline was associated with poorer baseline examination scores, but not with endline examination scores. This difference may be due to chance or due to the longer temporal gap between missing school at baseline and examination performance at endline. Persistent school absenteeism increases the risk of school dropout, lowers girls' self-esteem, and compromises future productivity [27]. A strength of our study was the use of an independent examination set by UNEB. We identified one previous study in Uganda which evaluated academic performance similarly, using assessments in English language, Mathematics, Integrated Science, and Social Studies set by UNEB [28]. In this small randomised controlled trial of 60 participants in two primary schools in Uganda, a menstrual health intervention (which included telling menstrual stories and playing games related to menstrual management) was associated with improved academic scores at 6 weeks after the intervention [28].

Overall, research on the effects of menstrual health on examination performance remains limited, with scarce causal evidence [5]. Several other different forms of school absenteeism due to sickness, domestic violence have been associated with academic performance [29] but these maybe hard to separate from menstrual related absenteeism. In our study, although we observed associations between menstrual health factors and examination performance, these findings are

modest and should be interpreted cautiously due to potential limitations such as residual confounding and limited sensitivity of educational measures.

### 4.4. Strengths and limitations of the study

Strengths of our study include the measurement of multiple dimensions of menstrual health; use of both cross-sectional and temporal analyses (to minimise reverse causality); measurement of reported number of school-days missed due to menstruation over a defined period; use of a self-completed questionnaire (to minimise social desirability bias); use of validated tools for menstrual care confidence and unmet practice needs; and assessment of examination performance using an examination set independently by UNEB, tailored to material that had been taught across all 60 schools.

Limitations include possible recall bias from retrospective, self-reported outcome of school attendance, residual confounding, and challenges related to the long duration (one year) between baseline menstrual experiences and endline outcomes. The examination performance outcome is objective but limited, especially at baseline, by the relatively sparse amount of material that had been taught between schools re-opening after COVID-9 in January 2022, and the assessment in March 2022. This resulted in a lack of discrimination between potentially low- and high-achieving students. The results are from two districts in Central Uganda, and we cannot be sure of generalizability to other settings. Future research with longer follow-up, larger sample sizes, more sensitive and diverse educational outcome measures, and designs that better isolate menstrual health effects from other educational influences is needed to fully understand the impact of menstrual health on examination performance.

### 4.5. Conclusion and recommendation

Multiple dimensions of menstrual health such as use of inadequate products, negative attitudes towards menstruation, having many unmet menstrual needs, menstrual pain and experience of menstrual-related teasing are strongly associated with school absenteeism and subsequent poor education performance among female students in Ugandan secondary schools, but the absolute effect size is small as menstrual-related absenteeism is infrequent. This small effect size is a plausible explanation for why the MENISCUS intervention improved menstrual health indicators, but had no impact on educational outcomes [14]. The gap between improving menstrual health versus achieving tangible gains in educational outcomes emphasizes the need for more tailored, multifaceted approaches that address not only menstrual health but also broader contextual barriers to enhance educational outcomes. Nevertheless, improving menstrual health remains a critical priority globally. Addressing this issue is essential not only for safeguarding physical and psychological well-being but also advancing gender equality and human rights issues.

## Supporting information

**S1 Table. Baseline menstrual-related exposures by endline school absenteeism for any reason.**
(DOCX)

**S1 File. MENISCUS Protocol Version v7.0 February 2024.**
(DOCX)

**S1 Checklist. STROBE_checklist_cohort.**
(DOCX)

## Acknowledgments

We would like to thank the staff, students, and communities of participating schools for their engagement with the study; the stakeholders from the Uganda Ministry of Education and Sports; and District Education Officers in Wakiso and Kalungu districts.

## Author contributions

**Conceptualization:** Christopher Baleke, Kate A. Nelson, Katherine A. Thomas, Helen A. Weiss.

**Formal analysis:** Christopher Baleke, Levicatus Mugenyi, Helen A. Weiss.

**Funding acquisition:** Helen A. Weiss.

**Investigation:** Denis Ndekezi, Jonathan Reuben Enomut.

**Methodology:** Christopher Baleke, Levicatus Mugenyi, Kate A. Nelson, Helen A. Weiss.

**Supervision:** Helen A. Weiss.

**Writing – original draft:** Christopher Baleke, Helen A. Weiss.

**Writing – review & editing:** Kate A. Nelson, Katherine A. Thomas, Denis Ndekezi, Jonathan Reuben Enomut, Connie Alezuyo, John Jerrim.

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
