## [Decision Letter · Decision Letter 0]

31 Jul 2025

Associations of menstrual health with school absenteeism and educational performance among Ugandan secondary school students: A longitudinal study

PLOS ONE

Dear Dr. Baleke,

Thank you for submitting your manuscript to PLOS ONE. After careful consideration, we feel that it has merit but does not fully meet PLOS ONE’s publication criteria as it currently stands. Therefore, we invite you to submit a revised version of the manuscript that addresses the points raised during the review process.

All the reviewers have provided extensive comments which need addressing, I look forward to receiving the revised manuscript.

We look forward to receiving your revised manuscript.

Kind regards,

Alison Parker

Academic Editor

PLOS ONE

Journal Requirements:

6. Please include captions for your Supporting Information files at the end of your manuscript, and update any in-text citations to match accordingly. Please see our Supporting Information guidelines for more information: http://journals.plos.org/plosone/s/supporting-information .

7. Please upload a copy of your study protocol that was approved by your ethics committee/IRB as a Supporting Information file. By the study protocol, we mean the complete and detailed plan for the conduct and analysis of the trial approved by the ethics committee/IRB. Please send this in the original language. If this is in a language other than English, please also provide a translation. [https://journals.plos.org/plosone/s/submission-guidelines#loc-guidelines-for-specific-study-types

Reviewers' comments:

Reviewer's Responses to Questions

**Comments to the Author**

1. Is the manuscript technically sound, and do the data support the conclusions?

Reviewer #1: Partly

Reviewer #2: Partly

Reviewer #3: Yes

Reviewer #4: Yes

Reviewer #5: Partly

2. Has the statistical analysis been performed appropriately and rigorously?

Reviewer #1: No

Reviewer #2: Yes

Reviewer #3: Yes

Reviewer #4: Yes

Reviewer #5: N/A

3. Have the authors made all data underlying the findings in their manuscript fully available?

Reviewer #1: Yes

Reviewer #2: Yes

Reviewer #3: Yes

Reviewer #4: Yes

Reviewer #5: Yes

4. Is the manuscript presented in an intelligible fashion and written in standard English?

Reviewer #1: Yes

Reviewer #2: Yes

Reviewer #3: Yes

Reviewer #4: Yes

Reviewer #5: Yes

Reviewer #1: The paper lacks clarity from the statistical perspective. The statistical strategy is poorly presented in this study. Section 10 of the supplemental protocol is well written with sample size justification and various analysis options per endpoints discussed.

When one starts reading the paper, the design and parallel arms not even introduced in the Abstract. One has to wait until section 2.2 and the investigators finally state that the trial was a parallel-arm, cluster-randomized controlled trial in 60 schools with schools randomized 1:1 to either immediate or delayed intervention delivery.

1. What is the control arm mentioned in the abstract?

2. Also, the actual longitudinal input is not clear to the reader. Please relate immediate, delayed, and endline in a longitudinal context with respect to Figure 1.

3. In the Conclusion the authors note that among Ugandan students, multiple dimensions of menstrual health are associated with school absenteeism and educational performance. Where is the analysis of immediate or delayed intervention or is that not part of this report?

4 It looks like Tables 2 to 4 are baseline and Table 5 are endline results from the longitudinal approach? Is that correct?

5. Some sentences are vague. For example ,please explain line 292, ’ After adjusting for factors at the same or more distal levels, there was strong evidence that menstrual-related absenteeism and poorer examination performance were both associated…etc’. What does 'distal level' mean? Please be exact when you mention adjustment throughout the presentation.

6. The entire presentation has to be edited for organization and clarity.

Reviewer #2: 1. Abstract

• Clearly explain how factors such as inadequate menstrual materials, negative attitudes toward menstruation, unmet menstrual practice needs, and experiences of menstrual teasing influence educational performance, not just absenteeism. Also clarify what is meant by “multiple dimensions of menstrual health” and how these are associated with both absenteeism and educational performance(may be in the body).

• Article type: Why is this described as a clinical trial? Since the study used secondary data and so sign of any medical test. Please justify the article type or indicate this clearly in the methods section.

2. Introduction

• Citations: Use square brackets for in-text citations instead of parentheses, to maintain consistency with journal style.

• Objective: Ensure the objective gives equal emphasis to both class absenteeism and educational performance. This balance appears lacking in the last part of the introduction.

• MENISCUS: Define this acronym (e.g., “MENISCUS stands for...”) before using it, to help readers unfamiliar with the term.

3. Methods

• Government vs private schools: What was the purpose of sampling both government and private schools if the outcomes are not contrasted by school type? Please clarify.

• Interventions: Specify the interventions received by the treatment schools versus what the control schools received. One of the selection criteria mentioned was the presence of WASH services, suggesting there might not have been an intervention. Please clarify this point. If there were interventions, how they were managed and how frequent they were?

• Quality control: What methods were employed to minimize recall bias in reporting class absenteeism?

• If the main results of MENISCUS have already been published, what new insights does this data analysis provide? Clearly, highlight the novel contributions/how different is this finding?

• Why did you use both cross-sectional and longitudinal designs? Are there sufficient indicators that support the longitudinal approach? It seems data were collected only at baseline, end line, and through student self-reports, with no evidence of continuous data collection.

4. Results

• Findings on the effects of menstrual health on educational performance are limited and not strongly demonstrated. Consider discussing this limitation.

General comments

• Clearly distinguish between this study and the original MENISCUS study to avoid confusion.

• Explicitly state the specific menstrual-related factors analyzed, instead of repeatedly using the vague term “multiple dimensions.” This will make your findings clearer and more impactful.

• Follow PLOS ONE reference format

Reviewer #3: The authours have presented an intelligble fashion and the grammar is standard. The author should look at the punctuation and the tenses. The methodolgy is okay and the it is technically sound. The authour should avoid being too wordy. Some paragraphs are too long. He should make them brief and to the point.

Reviewer #4: Abstract

Overall the abstract needs a bit of work, context and clarification. Please provide a very brief (understanding the tight word count) description of the intervention in the methods (or background) to help orient the reader.

39: Do you mean “we analysed baseline and endline data”?

The results section needs tailoring.

47: At baseline 3312 participants reported menstruating in the last 6 months.

3312 of how many? Or are you trying to say that your sample was the 3312 participants who reported menstruating in the last 6 months. Please clarify or consider revising: “Of the 3312 participants who reported menstruating in the past 6-months at baseline, 323 (9.8%) reported missing at least one day of school per month due to menstruation.”

49: 1286 (38.8%) missed at least one day a month for any reason.

Is this relevant? Is this key to be reported in the abstract? Without context it lacks definition of why it is relevant.

50: Of the 1192 participants in the control arm seen at endline, 135 (11.3%) missed at 51 least one day due to menstruation

Why is the control endline reported, and not the intervention arm endline? I don’t understand the relevance of reporting the control endline school absenteeism without the intervention data. What are you trying to say here? Can you add some clarifying text or otherwise provide some other data to clarify why this has been reported.

53: associated with multiple dimensions of menstrual health

Multiple dimensions of poor menstrual health. The bracketed examples are the inverse of the dimensions on the conceptual model (described later). Potentially refer to the model here, saying “as per a conceptual model described in the paper” - or something to orient a reader who is not familiar with the model. Another way might be to use the phrasing as per the introduction “aspects of menstrual experience”. Perhaps instead of brackets, change to such as...

55: Menstrual teasing

Please use term menstrual-related teasing. “Menstrual teasing” is not correct (and not used consistently throughout paper). Note: Line 306: “menstrual teasing” and Line 324 should be edited for consistency.

Introduction

71: A widely-cited statistic that “one in 10 school-age girls in Africa misses school or drops out for reasons related to her period” is not evidence-based (3, 4).

I get what you are trying to say here, but it reads strangely to me. Consider rephrasing. “One in ten school-age girls in Africa missed school or drops out for reasons related to her period” is a widely –cited statistic that lacks scientific backing.

74: Consider: drop female – redundant. An average of 31%?

75: Edit: Far too long sentence. Use full-stop after longer duration, then begin sentence with Additionally, the review found menstruation also affected...

89: Consider changing knowledge to evidence or data.

96: LMICs

99: Change to: ...with academic performance a reported consequence of menstruation, mainly due to menstrual pain or heavy menstrual bleeding.

102: Broader educational issues are written here as self-esteem, school participation, and educational performance.

How is educational performance difference from academic performance? Above has been discussing academic performance, please clarify how or if educational performance is different and thus belongs to “broader educational issues.”

102/103: Should this sentence: “In the review among university students...”come above the sentence “There is relatively little and inconclusive evidence..” Im confused as to why this sentence separates the following comment on the review. Please edit or justify.

109: You introduce us to: “Our recent cluster-randomised trial” is this the same cluster-randomised trail described in the methods? Aka the MENISCUS? If so, introduce it here not below. Please clarify or change the use of “our.” This is confusing.

111: Add: showed strong evidence of an intervention effect on multiple dimensions of menstrual health, such as....

Please detail what dimensions were effected, considering below you say little on school absence and nothing on examination performance, or school absence overall.

Additionally, is examination performace to be understood as different to academic performance, or educational performance. Please standardize or define.

117- 121: I suggest moving this last introductory paragraph to the top of the introduction to orient the reader and introduce MENISCUS.

After the first sentence introducing the MENISCUS trial, please provide a brief line to describe what the trial is.

The last sentence in this paragraph has a typo. Do you mean “add to the knowledge-base" or “add to the existing evidence on the impact of..”

I would also advise revising the sentence to read: “based on the impact of multiple dimensions of menstrual health..” or “based on the impact of menstruation..”

Methods

124: Add “The MENSISCUS trial/study”

132: Be consistent with abstract ie: baseline and endline.

132: This should not be the first time you are spelling out the acronym for MENISCUS. Please introduce in the introduction as suggested and spell out on first use.

142-143: Decapitalise Education Institutions

143-148: This sentence is too long. Full-stop at 146 and then start sentence with “School enrolment was to be estimated as....”

148: When you say “we” here, I am confused if you are talking about MENISCUS or your analysis of the trial? There needs to be clearer description of how what you are doing is related to MENISCUS and if you referring to it as “yours” etc.

151: Provide explanation of Secondary 2 in parenthesis as above for School 1.

166-167: Decapitalise subject names (Mathematics, Biology, English Language) (English remains capitalised- but not language)

187: Different reference style

188: Here you use the term “Menstrual-related factors”. I am wondering if you mean the same when you talk about “dimensions of menstrual health or “aspects of menstrual experience.”

If so, could you please choose one, define it and standardize throughout the paper.

190: Delete full-stop after confounders.

198: See 188

Results:

242: Describe/contextualise MPSN score. Is 2.11positive/negative?

Discussion:

355: “These results add..”

365: Support – remove s

378: Sentence need clarification - “but reported that other girls were absent from school due to menstruation” or “but reported that other girls reported absenteeism”?

381-382: Perhaps belongs in strengths and limitations

386: Suggest without adequate (not with inadequate).

392: Menstrual preparedness. Where is this definition of menstrual preparedness from? I don't see obtaining, using, cleaning and disposing of menstrual products and seeking assistance when needed as menstrual preparedness.

Menstrual preparedness is yes about tracking cycle and body literacy. Or pre-menarche about prior knowledge, education and support.

Reviewer #5: This is an automated report for PONE-D-25-29440. This report was solicited by the PLOS One editorial team and provided by ScreenIT.

ScreenIT is an independent group of scientists developing automated tools that analyze academic papers. A set of automated tools screened your submitted manuscript and provided the report below. Each tool was created by your academic colleagues with the goal of helping authors. The tools look for factors that are important for transparency, rigor and reproducibility, and we hope that the report might help you to improve reporting in your manuscript. Within the report you will find links to more information about the items that the tools check. These links include helpful papers, websites, or videos that explain why the item is important. While our screening tools aim to improve and maintain quality standards they may, on occasion, miss nuances specific to your study type or flag something incorrectly. Each tool has limitations that are described on the ScreenIT website. The tools screen the main file for the paper; they are not able to screen supplements stored in separate files. Please note that the Academic Editor had access to these comments while making a decision on your manuscript. The Academic Editor may ask that issues flagged in this report be addressed. If you would like to provide feedback on the ScreenIT tool, please email the team at ScreenIt@bih-charite.de. If you have questions or concerns about the review process, please contact the PLOS One office at plosone@plos.org.

**Do you want your identity to be public for this peer review?** For information about this choice, including consent withdrawal, please see our Privacy Policy

Reviewer #1: No

Reviewer #2: **Yes:** Andargie Fisseha

Reviewer #3: No

Reviewer #4: No

Reviewer #5: No

---

## [Author Response · Author response to Decision Letter 1]

9 Oct 2025

Dear Editor

Thank you for reviewing the manuscript. We have revised our manuscript to address the Editor and reviewer comments. Our detailed responses are below.

Response: We have followed these style requirements

Response: The funding statement has been removed from manuscript and is only in the Funding Statement.

Response: We confirm that the data are freely accessible upon request at the LSHTM Data Compass repository (MENISCUS Trial quantitative survey data) line 518-522

Response: Thank you for the guidance. The ethic statement has been moved to the Methods section (lines 238-242)

Response: We have two figures and provide captions in the manuscript (lines 233 and lines 258 respectively).

6. Please include captions for your Supporting Information files at the end of your manuscript, and update any in-text citations to match accordingly. Please see our Supporting Information guidelines for more information:

Response: We have added the caption for the Supporting Information file at the end of the manuscript (line 640) and its in-text citation has been updated on lines 333 and line 357

7. Please upload a copy of your study protocol that was approved by your ethics committee/IRB as a Supporting Information file. By the study protocol, we mean the complete and detailed plan for the conduct and analysis of the trial approved by the ethics committee/IRB. Please send this in the original language. If this is in a language other than English, please also provide a translation.

Response: The protocol has been uploaded as a supporting file

Response: There were no such comments

Reviewer Comments to the Author

Reviewer #1:

The paper lacks clarity from the statistical perspective. The statistical strategy is poorly presented in this study. Section 10 of the supplemental protocol is well written with sample size justification and various analysis options per endpoints discussed.

Response: Thank you for this comment. We have substantially revised the section on statistical analysis (Section 2.7) to clarify how we followed a conceptual framework (Figure 1) to build the multivariable models. Specifically, this first describes the two types of models used (mixed-effect negative binomial for school absenteeism, and mixed-effects linear regression for examination score) and then describes the conceptual framework and how the multivariable models were built for each level.

As this study was nested within the MENISCUS cluster-randomised controlled trial, we did not conduct a sample size calculation for this nested analysis.

When one starts reading the paper, the design and parallel arms not even introduced in the Abstract. One has to wait until section 2.2 and the investigators finally state that the trial was a parallel-arm, cluster-randomized controlled trial in 60 schools with schools randomized 1:1 to either immediate or delayed intervention delivery.

Response: We apologise for the confusion with the study design. We have clarified that this manuscript reports results from a secondary observational analysis using data from the published MENISCUS trial. It has been clarified in the abstract on lines 29-30, in the introduction on lines 114-119, and the methods on lines 130-144.

1. What is the control arm mentioned in the abstract?

Response: This is the control arm from the trial within which the current analyses are nested. We have clarified this in the abstract as follows “We conducted secondary observational analyses using data from a cluster-randomised trial of a menstrual health intervention in 60 Ugandan secondary schools. We used baseline data from trial participants in both arms, and endline data from the control arm participants. “(lines 29-30). We also now include a study flowchart (Figure 2) to help clarify this.

2. Also, the actual longitudinal input is not clear to the reader. Please relate immediate, delayed, and endline in a longitudinal context with respect to Figure 1.

Response: We thank the reviewer for this comment, and have removed the word ‘longitudinal’ throughout the manuscript. Instead, we describe the study as a prospective study in the title, and refer to baseline and endline data analysis in the abstract and manuscript (e.g. lines 29-30, 43-45 in the abstract), including in Figures 1 and the new Figure 2 which is the study flowchart.

3. In the Conclusion the authors note that among Ugandan students, multiple dimensions of menstrual health are associated with school absenteeism and educational performance. Where is the analysis of immediate or delayed intervention or is that not part of this report?

Response: Thank you for the comment. We have clarified in the Introduction section of the manuscript that this manuscript reports secondary data analysis of the trial, and that the intervention effects have been published previously (lines 103-113). We have also clarified that the estimate of the magnitude of effects in this paper help explain the lack of intervention effect (lines 380-383) “Through this observational analysis, we have shown that dimensions of menstrual health are associated with educational outcomes, but the absolute effect was small, which may help explain why the intervention was not sufficient to improve these educational outcomes.”

4 It looks like Tables 2 to 4 are baseline and Table 5 are endline results from the longitudinal approach? Is that correct?

Response: Yes that is correct. We apologise for an error in title of Table 5 which suggested the outcomes were at baseline (line 361).

5. Some sentences are vague. For example ,please explain line 292, ’ After adjusting for factors at the same or more distal levels, there was strong evidence that menstrual-related absenteeism and poorer examination performance were both associated…etc’. What does 'distal level' mean? Please be exact when you mention adjustment throughout the presentation.

Response: Thank you for this comment. We have revised the manuscript throughout to improve clarity. This includes rewriting Section 2.7 on Statistical Analyses to clarify the model building strategy . This includes definition of ‘distal level’ (to say “We use the term “distal” to refer to variables hypothesized to causally precede and influence variables at more proximal levels” (lines 209-220).

6. The entire presentation has to be edited for organization and clarity.

Response: We thank the reviewer for this comment and have edited the manuscript throughout to improve clarify.

Reviewer #2:

1. Abstract

• Clearly explain how factors such as inadequate menstrual materials, negative attitudes toward menstruation, unmet menstrual practice needs, and experiences of menstrual teasing influence educational performance, not just absenteeism.

Response: We had included this in the discussion but have moved to the introduction (lines 55-59).“Poor menstrual health is prevalent in low- and middle-income countries (LMICs), and may affect educational outcomes such as school absenteeism and examination performance through multiple pathways, including through shame, anxiety, lack of confidence, behavioural expectations, poor physical environments, and inadequate menstrual products”

Also clarify what is meant by “multiple dimensions of menstrual health” and how these are associated with both absenteeism and educational performance (may be in the body).

Response: We have defined the dimensions of menstrual health in the introduction (lines 70-73) “These dimensions include physical, behavioural and social components such as inadequate menstrual materials, menstrual pain, poor menstrual knowledge and confidence, and inadequate infrastructure for managing menstruation in schools”

• Article type: Why is this described as a clinical trial? Since the study used secondary data and so sign of any medical test. Please justify the article type or indicate this clearly in the methods section.

Response: Thank you for the comment. We have clarified in the manuscript that the paper is a secondary analysis using data from a cluster-randomised controlled trial (lines 123-124) and corrected this in the submission form.

2. Introduction

• Citations: Use square brackets for in-text citations instead of parentheses, to maintain consistency with journal style.

Response: Thank you. These have been changed to square brackets in line with journal style.

• Objective: Ensure the objective gives equal emphasis to both class absenteeism and educational performance. This balance appears lacking in the last part of the introduction. Response: We have revised the objective to include “examination performance” (line 114-117) and have revised the manuscript throughout to improve the balance.

• MENISCUS: Define this acronym (e.g., “MENISCUS stands for...”) before using it, to help readers unfamiliar with the term.

Response: We have defined “MENISCUS” as “Menstrual health interventions, schooling, and mental health problems among Ugandan students (MENISCUS)” at first use on lines 112-113.

3. Methods

• Government vs private schools: What was the purpose of sampling both government and private schools if the outcomes are not contrasted by school type? Please clarify.

Response: We have explained reasons for sampling both government and private schools on lines 156-158: “We over-sampled government schools as these are of interest to Ministry of Education stakeholders, and to increase generalisability of findings to these schools”.

• Interventions: Specify the interventions received by the treatment schools versus what the control schools received.

Response: We have added brief details of the intervention components for completeness on lines 137-140. We have also clarified that control schools were provided with a copy of the government guidelines on menstrual hygiene management and sexuality education and menstrual management reader.

One of the selection criteria mentioned was the presence of WASH services, suggesting there might not have been an intervention. Please clarify this point. If there were interventions, how they were managed and how frequent they were?

Response: We have clarified that we only use baseline trial data from the intervention arm in this paper, plus endline data from the control arm (lines 131-135) so there are no post-intervention data from the intervention arm. The WASH services mentioned were already present at the schools at baseline, and having “a minimum WASH facility” was one of the trial eligibility criteria. We have clarified this on lines 148-155.

• Quality control: What methods were employed to minimize recall bias in reporting class absenteeism?

Response: We have revised the methods to clarify “To minimize recall bias, we used cognitive testing before the survey to ensure clearly worded questions anchored to specific time frames, and provided reminders of school terms and holidays during survey completion. During self-completion of the questionnaires, research assistants were present to assist participants with clarifications about the recall period and absence reasons.” Lines 192-196

• If the main results of MENISCUS have already been published, what new insights does this data analysis provide? Clearly, highlight the novel contributions/how different is this finding?

Response: We have revised the discussion to highlight the new insights provided by this analysis (lines 376-382). “Through this observational analysis, we have shown that multiple dimensions of menstrual health are associated with educational outcomes, but the absolute effect was small, which may explain why the intervention was not sufficient to improve these educational outcomes”.

• Why did you use both cross-sectional and longitudinal designs? Are there sufficient indicators that support the longitudinal approach? It seems data were collected only at baseline, end line, and through student self-reports, with no evidence of continuous data collection.

Response: We chose to analyse prospective data in the control arm to address the possibility of reverse causality which could be in the baseline cross-sectional data. For example, it is possible that students with more absenteeism missed learning about how to manage their menstruation, and had poorer menstrual experiences. We chose to restrict this analysis to the control arm to assess this relationship in the absence of the MENISCUS intervention and hence make the results more generalisable. We have clarified this in the methods section (lines 227-231).

“To mitigate against the possibility of reverse causality, and to assess temporal associations of exposures with the outcomes, we also analysed associations of baseline exposures with endline outcomes in the control arm, using analogous methods. We restricted these analyses to participants in the control arm as the MENISCUS intervention may have modified the association.”

4.

Results

• Findings on the effects of menstrual health on educational performance are limited and not strongly demonstrated. Consider discussing this limitation.

Response: Thank you for your comment. We have improved the discussion for example line 434-436 “However, overall, associations of dimensions of menstrual health with examination scores were weaker than with school absenteeism. This is as expected given that absenteeism is more proximal to poor menstrual health than educational performance”

We have revised the limitations section to note that these findings are modest and should be interpreted cautiously due to potential limitations such as residual confounding and limited sensitivity of educational measures (Lines 458-462). Also future research with longer follow-up, larger sample sizes, more sensitive and diverse academic outcome measures, and designs that better isolate menstrual health effects from other educational influences is needed to fully understand the impact of menstrual health on academic achievement (lines 479-482)

General comments

• Clearly distinguish between this study and the original MENISCUS study to avoid confusion.

Response: Thank you for the comment. We have revised the paper throughout to clarify that this paper u

---

## [Decision Letter · Decision Letter 1]

26 Nov 2025

Dear Dr. Baleke,

**There are a few more minor comments from reviewer 3 to address.**

We look forward to receiving your revised manuscript.

Kind regards,

Alison Parker

Academic Editor

PLOS ONE

**Journal Requirements:**

Reviewers' comments:

Reviewer's Responses to Questions

**Comments to the Author**

Reviewer #1: All comments have been addressed

Reviewer #2: All comments have been addressed

Reviewer #3: All comments have been addressed

2. Is the manuscript technically sound, and do the data support the conclusions?

Reviewer #1: (No Response)

Reviewer #2: Yes

Reviewer #3: Yes

3. Has the statistical analysis been performed appropriately and rigorously?

Reviewer #1: (No Response)

Reviewer #2: Yes

Reviewer #3: N/A

4. Have the authors made all data underlying the findings in their manuscript fully available?

Reviewer #1: (No Response)

Reviewer #2: Yes

Reviewer #3: Yes

5. Is the manuscript presented in an intelligible fashion and written in standard English?

Reviewer #1: (No Response)

Reviewer #2: Yes

Reviewer #3: Yes

**Reviewer #1:** Changing the format to prospective and not longitudinal helped. New Figure 2 added to the clarity. Explanations and revisions were noted by the track changes.

**Reviewer #2:**  I have gone through the comments I provided to the authors' during the first review and therefore, I have found that the author's have addressed the comments adequately.

**Reviewer #3:**  The grammatical errors that I sited have been addressed. The author can make the paragraphs brief and cite appropriately

**Do you want your identity to be public for this peer review?** For information about this choice, including consent withdrawal, please see our Privacy Policy

Reviewer #1: No

Reviewer #2: **Yes:** Fisseha Atale

Reviewer #3: No

---

## [Author Response · Author response to Decision Letter 2]

2 Dec 2025

Dear Editor

Thank you for reviewing the manuscript. We have revised our manuscript to address the Editor and reviewer comments. Our detailed responses are below.

Reviewer #3: The grammatical errors that I sited have been addressed. The author can make the paragraphs brief and cite appropriately

We thank the reviewer for this comment. We have made paragraphs briefer (lines 103-111; 186-187; 264-265; 311-312; 319-320).

We have also added a citation on line 111 as requested.

---

## [Editor Report · Decision Letter 2]

10 Dec 2025

Associations of menstrual health with school absenteeism and examination performance among Ugandan secondary school students: A prospective study

PONE-D-25-29440R2

Dear Dr. Baleke,

We’re pleased to inform you that your manuscript has been judged scientifically suitable for publication and will be formally accepted for publication once it meets all outstanding technical requirements.

Kind regards,

Alison Parker

Academic Editor

PLOS ONE
---

## [Editor Report · Acceptance letter]

PONE-D-25-29440R2

PLOS One

Dear Dr. Baleke,

I'm pleased to inform you that your manuscript has been deemed suitable for publication in PLOS One. Congratulations! Your manuscript is now being handed over to our production team.

Kind regards,

on behalf of

Dr. Alison Parker

Academic Editor

PLOS One